# Unsupervised Object Segmentation by Redrawing

**Mickaël Chen**
Sorbonne Université, CNRS, LIP6, F-75005, Paris, France
mickael.chen@lip6.fr

**Thierry Artières**
Aix Marseille Univ, Université de Toulon, CNRS, LIS, Marseille, France
Ecole Centrale Marseille
thierry.artieres@centrale-marseille.fr

**Ludovic Denoyer**
Facebook Artificial Intelligence Research
denoyer@fb.com

## Abstract

Object segmentation is a crucial problem that is usually solved by using supervised learning approaches over very large datasets composed of both images and corresponding object masks. Since the masks have to be provided at pixel level, building such a dataset for any new domain can be very time-consuming. We present ReDO, a new model able to extract objects from images without any annotation in an unsupervised way. It relies on the idea that it should be possible to change the textures or colors of the objects without changing the overall distribution of the dataset. Following this assumption, our approach is based on an adversarial architecture where the generator is guided by an input sample: given an image, it extracts the object mask, then redraws a new object at the same location. The generator is controlled by a discriminator that ensures that the distribution of generated images is aligned to the original one. We experiment with this method on different datasets and demonstrate the good quality of extracted masks.

## 1 Introduction

Image segmentation aims at splitting a given image into a set of non-overlapping regions corresponding to the main components in the image. It has been studied for a long time in an unsupervised setting using prior knowledge on the nature of the region one wants to detect using e.g. normalized cuts and graph-based methods. Recently the rise of deep neural networks and their spectacular performances on many difficult computer vision tasks have led to revisit the image segmentation problem using deep networks in a fully supervised setting [5, 20, 58], a problem referred as semantic image segmentation.

Although such modern methods allowed learning successful semantic segmentation systems, their training requires large-scale labeled datasets with usually a need for pixel-level annotations. This feature limits the use of such techniques for many image segmentation tasks for which no such large scale supervision is available. To overcome this drawback, we follow here a very recent trend that aims at revisiting the unsupervised image segmentation problem with new tools and new ideas from the recent history and success of deep learning [55] and from the recent results of supervised semantic segmentation [5, 20, 58].

Building on the idea of scene composition [4, 14, 18, 56] and on the adversarial learning principle [17], we propose to address the unsupervised segmentation problem in a new way. We start by postulating an underlying generative process for images that relies on an assumption of independence between regions of an image we want to detect. This means that replacing one object in the image with another one, e.g. a generated one, should yield a realistic image. We use such a generative model as a backbone for designing an object segmentation model we call ReDO (ReDrawing of Objects), which outputs are then used to modify the input image by redrawing detected objects. Following ideas from adversarial learning, the supervision of the whole system is provided by a discriminator that is trained to distinguish between real images and fake images generated accordingly to the generative process. Despite being a simplified model for images, we find this generative process effective for learning a segmentation model.

The paper is organized as follows. We present related work in Section 2, then we describe our method in Section 3. We first define the underlying generative model that we consider in Section 3.2 and detail how we translate this hypothesis into a neural network architecture to learn a segmentation module in Section 3.3. Then we give implementation details in Section 4. Finally, we present experimental results on three datasets in Section 5 that explore the feasibility of unsupervised segmentation within our framework and compare its performance against a baseline supervised with few labeled examples.

## 2 Related Work

Image segmentation is a very active topic in deep learning that boasts impressive results when using large-scale labeled datasets. Those approaches can effectively parse high-resolution images depicting complex and diverse real-world scenes into informative semantics or instance maps. State-of-the-art methods use clever architectural choices or pipelines tailored to the challenges of the task [5, 20, 58].

However, most of those models use pixel-level supervision, which can be unavailable in some settings, or time-consuming to acquire in any case. Some works tackle this problem by using fewer labeled images or weaker overall supervision. One common strategy is to use image-level annotations to train a classifier from which class saliency maps can be obtained. Those saliency maps can then be exploited with other means to produce segmentation maps. For instance, WILDCAT [13] uses a Conditional Random Field (CRF) for spatial prediction in order to post-process class saliency maps for semantic segmentation. PRM [59], instead, finds pixels that provoke peaks in saliency maps and uses these as a reference to choose the best regions out of a large set of proposals previously obtained using MCG [2], an unsupervised region proposal algorithm. Both pipelines use a combination of a deep classifier and a method that take advantage of spatial and visual handcrafted image priors.

Co-segmentation, introduced by Rother et al. 2006 [46], addresses the related problem of segmenting objects that are shared by multiple images by looking for similar data patterns in all those images. Like the aforementioned models, in addition to prior image knowledge, deep saliency maps are often used to localize those objects [23]. Unsupervised co-segmentation [22], i.e. the task of covering objects of a specific category without additional data annotations, is a setup that resembles ours. However, unsupervised co-segmentation systems are built on the idea of exploiting features similarity and can't easily be extended to a class-agnostic system. As we aim to ultimately be able to segment very different objects, our approach instead relies on independence between the contents of different regions of an image which is a more general concept.

Fully unsupervised approaches have traditionally been more focused on designing handcrafted features or energy functions to define the desired property of *objectness*. Impressive results have been obtained when making full use of depth maps in addition to usual RGB images [44, 49] but it is much harder to specify good energy functions for purely RGB images. W-NET [55] extracts latent representations via a deep auto-encoder that can then be used by a more classic CRF algorithm. Kanezaki 2018 [28] further incorporate deep priors and train a neural network to directly minimize their chosen intra-region pixel distance. A different approach is proposed by Ji et al. 2019 [26] whose method finds clusters of pixels using a learned distance invariant to some known properties. Unlike ours, none of these approaches are learned entirely from data.

Our work instead follows a more recent trend by inferring scene decomposition directly from data. Stemming from DRAW [19], many of those approaches [4, 14] use an attention network to read a region of an image and a Variational Auto-encoder (VAE) to partially reconstruct the image in an iterative process in order to flesh out a meaningful decomposition. LR-GAN [56] is able to generate

simple scenes recursively, building object after object, and Sbai et al. 2018 [48] decompose an image into single-colored strokes for vector graphics. While iterative processes have the advantage of being able to handle an arbitrary number of objects, they are also more unstable and difficult to train. Most of those can either only be used in generation [56], or only handle very simple objects [4, 14, 18]. As a proof of concept, we decided to first ignore this additional difficulty by only handling a set number of objects but our model can naturally be extended with an iterative composition process. This choice is common among works that, like ours, focus on other aspects of image compositionality. Van Steenkiste et al. 2018 [52] advocates for a generative framework that accounts for relationship between objects. While they do produce masks as part of their generative process, they cannot segment a given image. Closer to our setup, the very recent IODINE [18] propose a VAE adapted for multi-objects representations. Their learned representations include a scene decomposition, but they need a costly iterative refinement process whose performance have only been demonstrated on simulated datasets and not real images. Like ours, some prior work have tried to find segmentation mask by recomposing new images. SEIGAN [42] and Cut & Paste [45] learns to separate object and background by moving the region corresponding to the object to another background and making sure the image is still realistic. These methods however, need to have access to background images without objects, which might not be easy to obtain.

Our work also ties to recent research in disentangled representation learning. Multiple techniques have been used to separate information in factored latent representations. One line of work focuses on understanding and exploiting the innate disentangling properties of Variational Auto-Encoders. It was first observed by $\beta$-VAE [21] that VAEs can be constrained to produce disentangled representations by imposing a stronger penalty on the Kullback-Leibler divergence term on the VAE loss. FactorVAE [30] and $\beta$-TCVAE [7] extract a total correlation term from the KL term of the VAE objective and specifically re-weight it instead of the whole KL term. In a similar fashion, HFVAE [15] introduces a hierarchical decomposition of the KL term to impose a structure on the latent space. A similar property can be observed with GAN-based models, as shown by InfoGAN [9] which forces a generator to map a code to interpretable features by maximizing the mutual information between the code and the output. Using adversarial training is also a good way to split and control information in latent embeddings. Fader Networks [32] uses adversarial training to remove specific class information from a vector. This technique is also used in adversarial domain adaptation [16, 36, 51] to align embeddings from different domains. Similar methods can be used to build factorial representations instead of simply removing information [6, 10, 11, 38]. Like our work, they use adversarial learning to match an implicitly predefined generative model but for purposes unrelated to segmentation.

## 3 Method

### 3.1 Overview

A segmentation process F splits a given image $\mathbf{I} \in \mathbb{R}^{W \times H \times C}$ into a set of non-overlapping regions. F can be described as a function that assigns to each pixel coordinate of $\mathbf{I}$ one of $n$ regions. The problem is then to find a correct partition F for any given image $\mathbf{I}$. Lacking supervision, a common strategy is to define properties one wants the regions to have, and then to find a partition that produces regions with such properties. This can be done by defining an energy function and then finding an optimal split. The challenge is then to accurately describe and model the statistical properties of meaningful regions as a function one can optimize.

We address this problem differently. Instead of trying to define the right properties of regions at the level of each image, we make assumptions about the underlying generative process of images in which the different regions are explicitly modeled. Then, by using an adversarial approach, we learn the parameters of the different components of our model so that the overall distribution of the generated images matches the distribution of the dataset. We detail the generative process in the section 3.2, while the way we learn F is detailed in Section 3.3.

### 3.2 Generative Process

We consider that images are produced by a generative process that operates in three steps: first, it defines the different regions in the image i.e the organization of the scene (*composition step*). Then, given this segmentation, the process generates the pixels for each of the regions **independently** (*drawing step*). At last, the resulting regions are assembled into the final image (*assembling step*).

Let us consider a scene composed of $n-1$ objects and one background we refer to as object $n$. Let us denote $\mathbf{M}^k \in \{0,1\}^{W \times H}$ the mask corresponding to object $k$ which associates one binary value to each pixels in the final image so that $\mathbf{M}^k_{x,y} = 1$ iff the pixel of coordinate $(x,y)$ belongs to object $k$. Note that, since one pixel can only belong to one object, the masks have to satisfy $\sum_{k=1}^{n} \mathbf{M}^k_{x,y} = 1$ and the background mask $\mathbf{M}^n$ can therefore easily be retrieved computed from the object masks as $\mathbf{M}^n = 1 - \sum_{k=1}^{n-1} \mathbf{M}^k$.

The pixel values of each object $k$ are denoted $\mathbf{V}^k \in \mathbb{R}^{W \times H \times C}$. Given that the image we generate is of size $W \times H \times C$, each object is associated with an image of the same size but only the pixels selected by the mask will be used to compose the output image. The final composition of the objects into an image is computed as follows:

$$\mathbf{I} \leftarrow \sum_{k=1}^{n} \mathbf{M}^k \odot \mathbf{V}^k.$$

To recap, the underlying generative process described previously can be summarized as follow: i) first, the masks $\mathbf{M}^k$ are chosen together based on a mask prior $p(\mathbf{M})$. ii) Then, for each object independently, the pixel values are chosen based on a distribution $p(\mathbf{V}^k|\mathbf{M}^k, k)$. iii) Finally, the objects are assembled into a complete image.

This process makes an assumption of independence between the colors and textures of the different objects composing a scene. While this is a naive assumption, as colorimetric values such as exposition, brightness, or even the real colors of two objects, are often related, this simple model still serves as a good prior for our purposes.

### 3.3 From Generative Process to Object Segmentation

Now, instead of considering a purely generative process where the masks are generated following a prior $p(\mathbf{M})$, we consider the inductive process where the masks are extracted directly from any input image $\mathbf{I}$ through the function $\mathtt{F}$ which is the object segmentation function described previously. The role of $\mathtt{F}$ is thus to output a set of masks given any input $\mathbf{I}$. The new generative process acts as follows: i) it takes a random image in the dataset and computes the masks using $\mathtt{F}(\mathbf{I}) \rightarrow \mathbf{M}_1, \ldots, \mathbf{M}_n$, and ii) it generates new pixel values for the regions in the image according to a distribution $p(\mathbf{V}^k|\mathbf{M}^k, k)$. iii) It aggregates the objects as before.

In order for output images to match the distribution of the training dataset, all the components (i.e $\mathtt{F}$ and $p(\mathbf{V}^k|\mathbf{M}^k, k)$) are learned adversarially following the GAN approach. Let us define $\mathtt{D}: \mathbb{R}^{W \times H \times C} \rightarrow \mathbb{R}$ a discriminator function able to classify images as *fake* or *real*. Let us denote $\mathtt{G}_\mathtt{F}(\mathbf{I}, \mathbf{z}_1, \ldots, \mathbf{z}_n)$ our generator function able to compose a new image given an input image $\mathbf{I}$, an object segmentation function $\mathtt{F}$, and a set of vectors $\mathbf{z}_1, \ldots, \mathbf{z}_n$ each sampled independently following a prior $p(\mathbf{z})$ for each object $k$, background included. Since the pixel values of the different regions are considered as independent given the segmentation, our generator can be decomposed in $n$ generators denoted $\mathtt{G}_k(\mathbf{M}^k, \mathbf{z}_k)$, each one being in charge of deciding the pixel values for one specific region. The complete image generation process thus operates in three steps:

$$1)\ \mathbf{M}^1, \ldots, \mathbf{M}^n \leftarrow \mathtt{F}(\mathbf{I}) \qquad \text{(composition step)}$$

$$2)\ \mathbf{V}^k \leftarrow \mathtt{G}_k(\mathbf{M}^k, \mathbf{z}_k)\ \text{for } k \in \{1, \ldots, n\} \qquad \text{(drawing step)}$$

$$3)\ \mathtt{G}_\mathtt{F}(\mathbf{I}, \mathbf{z}_1, \ldots, \mathbf{z}_n) = \sum_{k=1}^{n} \mathbf{M}^k \odot \mathbf{V}^k \qquad \text{(assembling step)}.$$

Provided the functions $\mathtt{F}$ and $\mathtt{G}_k$ are differentiable, they can thus be learned by solving the following adversarial problem:

$$\min_{\mathtt{G}_\mathtt{F}} \max_{\mathtt{D}} \mathcal{L} = \mathbb{E}_{\mathbf{I} \sim p_{data}}\big[\log \mathtt{D}(\mathbf{I})\big] + \mathbb{E}_{\mathbf{I} \sim p_{data}, \mathbf{z}_1, \ldots \mathbf{z}_n \sim p(\mathbf{z})}\big[\log(1 - \mathtt{D}(\mathtt{G}_\mathtt{F}(\mathbf{I}, \mathbf{z}_1, \ldots, \mathbf{z}_n)))\big].$$

Therefore, in practice we have $\mathtt{F}$ output soft masks in $[0, 1]$ instead of binary masks. Also, in line with recent GAN literature [3, 39, 50, 57], we choose to use the hinge version of the adversarial loss

[35, 50] instead, and obtain the following formulation:

$$\max_{\texttt{G}_\texttt{F}} \mathcal{L}_\texttt{G} = \mathbb{E}_{\mathbf{I}\sim p_{data},\mathbf{z}_1,\ldots,\mathbf{z}_n\sim p(\mathbf{z})}\big[\texttt{D}(\texttt{G}_\texttt{F}(\mathbf{I},\mathbf{z}_1,\ldots,\mathbf{z}_n))\big]$$

$$\max_{\texttt{D}} \mathcal{L}_\texttt{D} = \mathbb{E}_{\mathbf{I}\sim p_{data}}\big[\min(0,-1+\texttt{D}(\mathbf{I}))\big]$$
$$+ \mathbb{E}_{\mathbf{I}\sim p_{data},\mathbf{z}_1,\ldots,\mathbf{z}_n\sim p(\mathbf{z})}\big[\min(0,-1-\texttt{D}(\texttt{G}_\texttt{F}(\mathbf{I},\mathbf{z}_1,\ldots,\mathbf{z}_n)))\big].$$

Still, as it stands, the learning process of this model may fail for two reasons. First, it does not have to extract a meaningful segmentation in regards to the input $\mathbf{I}$. Indeed, since the values of all the output pixels will be generated, $\mathbf{I}$ can be ignored entirely to generate plausible pictures. For instance, the segmentation could be the same for all the inputs regardless of input $\mathbf{I}$. Second, it naturally converges to a trivial extractor $\texttt{F}$ that puts the whole image into a single region, the other regions being empty. We thus have to add additional constraints to our model.

**Constraining mask extraction by redrawing a single region.** The first constraint aims at forcing the model to extract meaningful region masks instead of ignoring the image. To this end, we take advantage of the assumption that the different objects are independently generated. We can, therefore, replace only one region at each iteration instead of regenerating all the regions. Since the generator now has to use original pixel values from the image in the reassembled image, it cannot make arbitrary splits. The generation process becomes as follows:

$$
\begin{aligned}
&1)\ \mathbf{M}^1,\ldots,\mathbf{M}^n \leftarrow \texttt{F}(\mathbf{I}) && \text{(composition step)}\\
&2)\ \mathbf{V}^k \leftarrow \mathbf{I} \text{ for } k\in\{1,\ldots,n\}\setminus\{\mathbf{i}\}\\
&\quad\ \mathbf{V}^\mathbf{i} \leftarrow \texttt{G}_\mathbf{i}(\mathbf{M}^\mathbf{i},\mathbf{z}_\mathbf{i}) && \text{(drawing step)}\\
&3)\ \texttt{G}_\texttt{F}(\mathbf{I},\mathbf{z}_\mathbf{i},\mathbf{i}) = \sum_{k=1}^{n}\mathbf{M}^k\odot\mathbf{V}^k && \text{(assembling step)},
\end{aligned}
$$

where $\mathbf{i}$ designates the index of the only region to redraw and is sampled from $\mathcal{U}(n)$, the discrete uniform distribution on $\{1,\ldots,n\}$. The new learning objectives are as follows:

$$\max_{\texttt{G}_\texttt{F}} \mathcal{L}_\texttt{G} = \mathbb{E}_{\mathbf{I}\sim p_{data},\mathbf{i}\sim\mathcal{U}(n),\mathbf{z}_\mathbf{i}\sim p(\mathbf{z})}\big[\texttt{D}(\texttt{G}_\texttt{F}(\mathbf{I},\mathbf{z}_\mathbf{i},\mathbf{i}))\big]$$

$$\max_{\texttt{D}} \mathcal{L}_\texttt{D} = \mathbb{E}_{\mathbf{I}\sim p_{data}}[\min(0,-1+\texttt{D}(\mathbf{I}))] + \mathbb{E}_{\mathbf{I}\sim p_{data},\mathbf{i}\sim\mathcal{U}(n),\mathbf{z}_\mathbf{i}\sim p(\mathbf{z})}[\min(0,-1-\texttt{D}(\texttt{G}_\texttt{F}(\mathbf{I},\mathbf{z}_\mathbf{i},\mathbf{i})))].$$

**Conservation of Region Information.** The second constraint is that given a region $\mathbf{i}$ generated from a latent vector $\mathbf{z}_\mathbf{i}$, the final image $\texttt{G}_\texttt{F}(\mathbf{I},\mathbf{z}_\mathbf{i},\mathbf{i})$ must contain information about $\mathbf{z}_\mathbf{i}$. This constraint is designed to prevent the mask extractor $\texttt{F}$ to produce empty regions. Indeed, if region $\mathbf{i}$ is empty, i.e. $\mathbf{M}^\mathbf{i}_{x,y}=0$ for all $x,y$, then $\mathbf{z}_\mathbf{i}$ cannot be retrieved from the final image. Equivalently, if $\mathbf{z}_\mathbf{i}$ can be retrieved, then region $\mathbf{i}$ is not empty. This information conservation constraint is implemented through an additional term in the loss function. Let us denote $\delta_k$ a function which objective is to infer the value of $\mathbf{z}_k$ given any image $\mathbf{I}$. One can learn such a function simultaneously to promote conservation of information by the generator. This strategy is similar to the mutual information maximization used in InfoGAN. [9].

The final complete process is illustrated in Figure 1 and correspond to the following learning objectives:

$$\max_{\texttt{G}_\texttt{F},\delta} \mathcal{L}_\texttt{G} = \mathbb{E}_{\mathbf{I}\sim p_{data},\mathbf{i}\sim\mathcal{U}(n),z_\mathbf{i}\sim p(z)}\big[\texttt{D}(\texttt{G}_\texttt{F}(\mathbf{I},\mathbf{z}_\mathbf{i},\mathbf{i})) - \lambda_z||\delta_\mathbf{i}(\texttt{G}_\texttt{F}(\mathbf{I},\mathbf{z}_\mathbf{i},\mathbf{i})) - \mathbf{z}_\mathbf{i}||_2^2\big]$$

$$\max_{\texttt{D}} \mathcal{L}_\texttt{D} = \mathbb{E}_{\mathbf{I}\sim p_{data}}\big[\min(0,-1+\texttt{D}(\mathbf{I})\big] + \mathbb{E}_{\mathbf{I}\sim p_{data},\mathbf{i}\sim\mathcal{U}(n),\mathbf{z}_\mathbf{i}\sim p(\mathbf{z})}\big[\min(0,-1-\texttt{D}(\texttt{G}_\texttt{F}(\mathbf{I},\mathbf{z}_\mathbf{i},\mathbf{i}))\big],$$

where $\lambda_z$ is a fixed hyper-parameter that controls the strength of the information conservation constraint. Note that the constraint is necessary for our model to find non trivial solutions, as otherwise, putting the whole image into a single region is both optimal and easy to discover for the neural networks. The final learning algorithm follows classical GAN schema [3, 17, 39, 57] by updating the generator and the discriminator alternatively with the update functions presented in Algorithm 1.

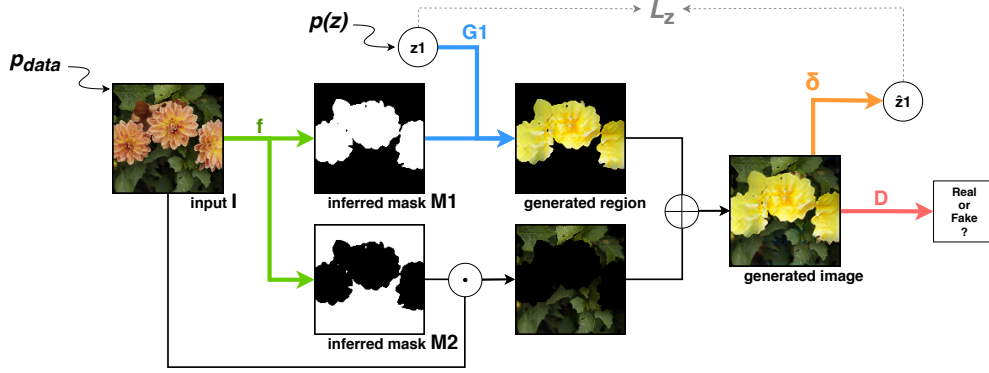

Figure 1: Example generation with $G_f(I, z_i, i)$ with $i = 1$ and $n = 2$. Learned functions are in color.

---

**Algorithm 1** Networks update functions

---
 1: **procedure** GENERATORUPDATE
 2:      sample data $I \sim p_{data}$,
 3:      sample region $i \sim \texttt{Uniform}(\{1, \ldots, n\})$
 4:      sample noise vector $z_i \sim p(z)$
 5:      $I_{gen} \leftarrow G_f(I, z_i, i)$                               ▷ generate image
 6:      $\mathcal{L}_z \leftarrow -\|\delta_i(I_{gen}) - z_i\|_2$         ▷ compute information conservation loss
 7:      $\mathcal{L}_G \leftarrow D(I_{gen})$                         ▷ compute adversarial loss
 8:      update $G_f$ with $\nabla_{G_f}\big[\mathcal{L}_G + \mathcal{L}_z\big]$
 9:      update $\delta_i$ with $\nabla_{\delta_i}\mathcal{L}_z$
10: **procedure** DISCRIMINATORUPDATE
11:      sample datapoints $I_{real}, I_{input} \sim p_{data}$
12:      sample region $i \sim \texttt{Uniform}(\{1, \ldots, n\})$
13:      sample noise vector $z_i \sim p(z)$
14:      $I_{gen} \leftarrow G_f(I_{input}, z_i, i)$                       ▷ generate image
15:      $\mathcal{L}_D \leftarrow \min(0, -1 + D(I_{real})) + \min(0, -1 - D(I_{gen})$  ▷ compute adversarial loss
16:      update $D$ with $\nabla_D\mathcal{L}_D$

---

# 4 Implementation

We now provide some information about the architecture of the different components (additional details are given in Supplementary materials). As usual with GAN-based methods, the choice of a good architecture is crucial. We have chosen to build on the GAN and the image segmentation literature and to take inspiration from the neural network architectures they propose.

For the mask generator F, we use an architecture inspired by PSPNet [58]. The proposed architecture is a fully convolutional neural network similar to one used in image-to-image translation [60], to which we add a Pyramid Pooling Module [58] whose goal is to gather information on different scales via pooling layers. The final representation of a given pixel is thus encouraged to contain local, regional, and global information at the same time.

The region generators $G_k$, the discriminator D and the network $\delta$ that reconstructs $z$ are based on SAGAN [57] that is frequently used in recent GAN literature [3, 37]. Notably, we use spectral normalization [39] for weight regularization for all networks except for the mask provider F, and we use self-attention [57] in $G_k$ and D to handle non-local relations. To both promote stochasticity in our generators and encourage our latent code $z$ to encode for texture and colors, we also use conditional batch-normalization in $G_k$. The technique has emerged from style modeling for style transfer tasks [12, 43] and has since been used for GANs as a mean to encode for style and to improve stochasticity [1, 8, 54]. All parameters of the different $\delta_k$ functions are shared except for their last layers.

As it is standard practice for GANs [3], we use orthogonal initialization [47] for our networks and ADAM [31] with $\beta = (0, .9)$ as optimizer. Learning rates are set to $10^{-4}$ except for the mask network F which uses a smaller value of $10^{-5}$. We sample noise vectors $\mathbf{z_i}$ of size 32 (except for MNIST where we used vectors of size 16) from $\mathcal{N}(0, I_d)$ distribution. We used mini-batches of size 25 and ran each experiment on a single NVidia Tesla P100 GPU. Despite our conservation of information loss, the model can still collapse into generating empty masks at the early steps of the training. While the regularization does alleviate the problem, we suppose that the mask generator F can collapse even before the network $\delta$ learns anything relevant and can act as a stabilizer. As the failures happen early and are easy to detect, we automatically restart the training should the case arise.

We identified $\lambda_z$ and the initialization scheme as critical hyper-parameters and focus our hyper-parameters search on those. More details, along with specifics of the implementation used in our experiments are provided as Supplementary materials. The code, dataset splits and pre-trained models are also available open-source [1].

## 5 Experiments

### 5.1 Datasets

We present results on three natural image datasets and one toy dataset. All images have been resized and then cropped to $128 \times 128$.

**Flowers** dataset [40, 41] is composed of 8189 images of flowers. The dataset is provided with a set of masks obtained via an automated method built specifically for flowers [40]. We split into sets of 6149 training images, 1020 validation and 1020 test images and use the provided masks as ground truth for evaluation purpose only.

**Labeled Faces in the Wild** dataset [25, 33] is a dataset of 13233 faces. A subpart of the funneled version [24] has been segmented and manually annotated [27], providing 2927 groundtruth masks. We use the non-annotated images for our training set. We split the annotated images between validation and testing sets so that there is no overlap in the identity of the persons between both sets. The test set is composed of 1600 images, and the validation set of 1327 images.

The **Caltech-UCSD Birds 200 2011 (CUB-200-2011)** dataset [53] is a dataset containing 11788 photographs of birds. We use 10000 images for our training split, 1000 for the test split, and the rest for validation.

As a sanity check, we also build a toy dataset **colored-2-MNIST** in which each sample is composed of an uniform background on which we draw two colored MNIST [34] numbers: one odd number and one even number. Odd and even numbers have colors sampled from different distributions so that our model can learn to differentiate them. For this dataset, we set $n = 3$ as there are three components.

As an additional experiment, we also build a new dataset by fusing Flowers and LFW datasets. This new **Flowers+LFW** dataset has more variability, and contains different type of objects. We used this dataset to demonstrate that ReDO can work without label information on problems with multiple categories of objects.

### 5.2 Results

To evaluate our method ReDO, we use two metrics commonly used for segmentation tasks. The pixel classification accuracy (Acc) measures the proportion of pixels that have been assigned to the correct region. The intersection over union (IoU) is the ratio between the area of the intersection between the inferred mask and the ground truth over the area of their union. In both cases, higher is better. Because ReDO is unsupervised and we can't control which output region corresponds to which object or background in the image, we compute our evaluation based on the regions permutation that matches the ground truth the best. For model selection, we used IoU computed on a held out labeled validation set. When available, we present our evaluation on both the training set and a test set as, in an unsupervised setting, both can be relevant depending on the specific use case. Results are presented in Table 1 and show that ReDO achieves reasonable performance on the three real-world datasets.

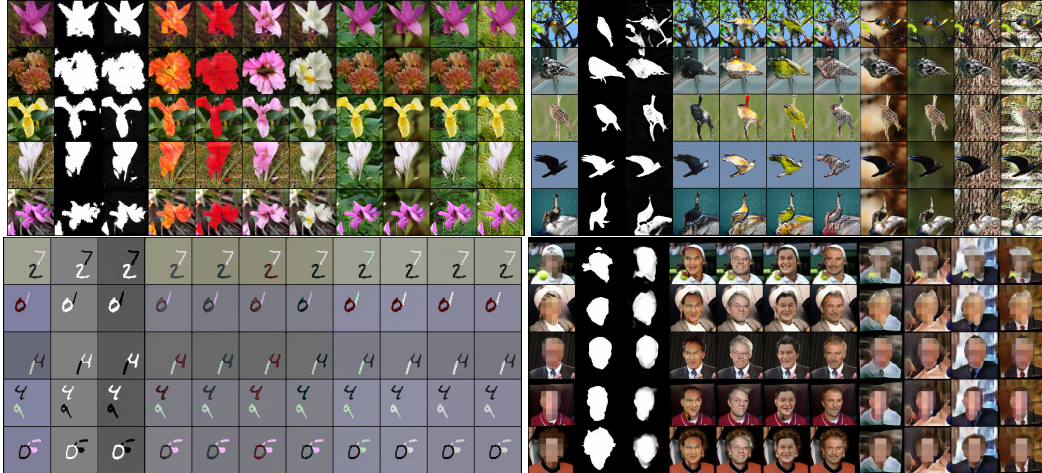

Figure 2: Generated samples (not cherry-picked, zoom in for better visibility). For each dataset, the columns are from left to right: 1) input images, 2) ground truth masks, 3) masks inferred by the model for object one, 4-7) generation by redrawing object one, 8-11) generation by redrawing object two. As we keep the same $z_i$ on any given column, the color and texture of the redrawn object is kept constant across rows. More samples are provided in Supplementary materials. Faces from the LFW dataset have been anonymized, in vizualisations only, to protect personality rights.

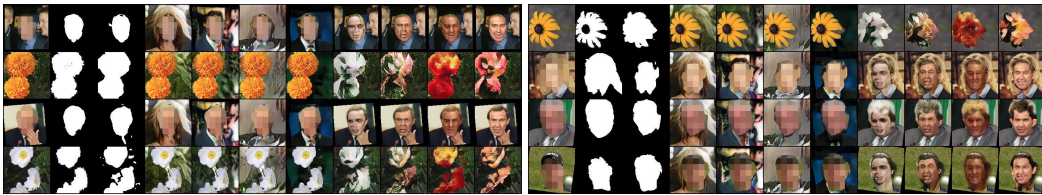

Figure 3: Results on LFW + Flowers dataset, arranged as in Figure 2. As $z$ is kept constant on a column across all rows, we can observe that $z$ codes for different textures depending on the class of the image even though the generator is never given this information explicitly. Faces from the LFW dataset have been anonymized, in vizualisations only, to protect personality rights.

We also compared the performance of ReDO, which is unsupervised, with a supervised method, keeping the same architecture for F in both cases. We analyze how many training samples are needed to reach the performance of the unsupervised model (see Figure 4). One can see that the unsupervised results are in the range of the ones obtained with a supervised method, and usually outperform supervised models trained with less than around 50 or 100 examples depending on the dataset. For instance, on the LFW Dataset, the unsupervised model obtains about 92% of accuracy and 79% IoU and the supervised model needs 50-60 labeled examples to reach similar performance.

We provide random samples of extracted masks (Figure 2) and the corresponding generated images with a redrawn object or background. Note that our objective is not to generate appealing images but to learn an object segmentation function. Therefore, ReDO generates images that are less realistic than the ones generated by state-of-the-art GANs. Focus is, instead, put on the extracted masks, and we can see the good quality of the obtained segmentation in many cases. Best and worst masks, as well as more random samples, are displayed in Supplementary materials.

We also trained ReDO on the fused Flowers+LFW dataset without labels. We re-used directly the hyper-parameters we have used to fit the Flowers dataset without further tuning and obtained, as preliminary results, a reasonable accuracy of 0.856 and an IoU of 0.691. This shows that ReDO is able to infer class information from masks even in a fully unsupervised setup. Samples are displayed in Figure 3.

| Dataset | Train Acc | Train IoU | Test Acc | Test IoU |
|---------|-----------|-----------|----------|----------|
| LFW | - | - | $0.917 \pm 0.002$ | $0.781 \pm 0.005$ |
| CUB | $0.840 \pm 0.012$ | $0.423 \pm 0.023$ | $0.845 \pm 0.012$ | $0.426 \pm 0.025$ |
| Flowers* | $0.886 \pm 0.008$ | $0.780 \pm 0.012$ | $0.879 \pm 0.008$ | $0.764 \pm 0.012$ |
| Flowers+LFW | - | - | 0.856 | 0.691 |

Table 1: Performance of ReDO in accuracy (Acc) and intersection over union (IoU) on retrieved masks. Means and standard deviations are based on five runs with fixed hyper-parameters. LWF train set scores are not available since we trained on unlabeled images. *Please note that segmentations provided along the original Flowers dataset [41] have been obtained using an automated method. We display samples with top disagreement masks between ReDO and ground truth in Supplementary materials. In those cases, we find ours to provide better masks.

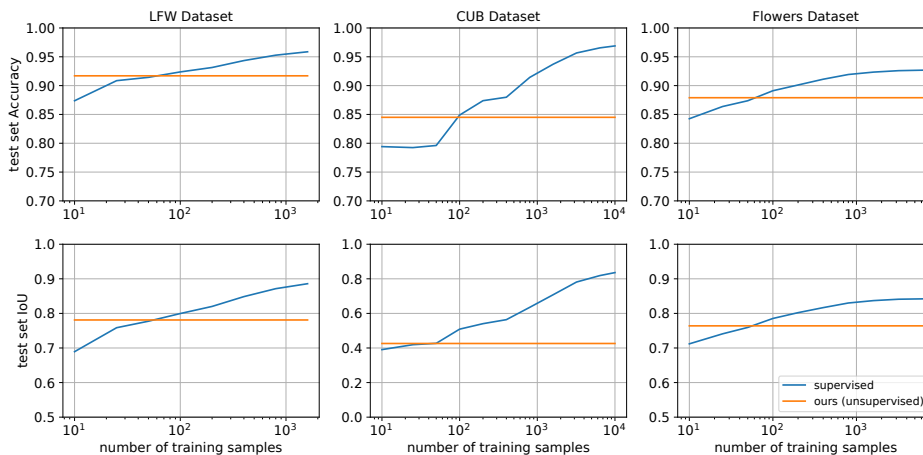

Figure 4: Comparison with supervised baseline as a function of the number of available training samples.

## 6 Conclusion

We presented a novel method called ReDO for unsupervised learning to segment images. Our proposal is based on the assumption that if a segmentation model is accurate, then one could edit any real image by replacing any segmented object in a scene by another one, randomly generated, and the result would still be a realistic image. This principle allows casting the unsupervised learning of image segmentation as an adversarial learning problem. Our experimental results obtained on three datasets show that this principle works. In particular, our segmentation model is competitive with supervised approaches trained on a few hundred labeled examples.

Our future work will focus on handling more complex and diverse scenes. As mentioned in Section 2, our model could generalize to an arbitrary number of objects and objects of unknown classes via iterative design and/or class agnostic generators. Currently, we are mostly limited by our ability to effectively train GANs on those more complicated settings but rapid advances in image generation [3, 29, 37] make it a reasonable goal to pursue in a near future. Meanwhile, we will be investigating the use of the model in a semi-supervised or weakly-supervised setup. Indeed, additional information would allow us to guide our model for harder datasets while requiring fewer labels than fully supervised approaches. Conversely, our model could act as a regularizer by providing a prior for any segmentation tasks.

## Acknowledgments

This work was supported by the French National Research Agency projects LIVES (grant number ANR-15-CE23-0026-03) and "Deep in France" (grant number ANR-16-CE23-0006).

## Footnotes

[1] https://github.com/mickaelChen/ReDO

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
