[Supplementary Material]

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

$$\mathbf{V}^{\mathbf{i}} \leftarrow \mathtt{G_i}(\mathbf{M}^{\mathbf{i}}, \mathbf{z_i}) \qquad \text{(drawing step)}$$
$$3)\ \mathtt{G_F}(\mathbf{I}, \mathbf{z_i}, \mathbf{i}) = \sum_{k=1}^{n} \mathbf{M}^k \odot \mathbf{V}^k \qquad \text{(assembling step)},$$

where $\mathbf{i}$ designates the index of the only region to redraw and is sampled from $\mathcal{U}(n)$, the discrete uniform distribution on $\{1, \ldots, n\}$. The new learning objectives are as follows:

$$\max_{\mathtt{G_F}} \mathcal{L}_{\mathtt{G}} = \mathbb{E}_{\mathbf{I}\sim p_{data}, \mathbf{i}\sim\mathcal{U}(n), \mathbf{z_i}\sim p(\mathbf{z})} \big[ \mathtt{D}(\mathtt{G_F}(\mathbf{I}, \mathbf{z_i}, \mathbf{i})) \big]$$
$$\max_{\mathtt{D}} \mathcal{L}_{\mathtt{D}} = \mathbb{E}_{\mathbf{I}\sim p_{data}} [\min(0, -1 + \mathtt{D}(\mathbf{I}))] + \mathbb{E}_{\mathbf{I}\sim p_{data}, \mathbf{i}\sim\mathcal{U}(n), \mathbf{z_i}\sim p(\mathbf{z})} [\min(0, -1 - \mathtt{D}(\mathtt{G_F}(\mathbf{I}, \mathbf{z_i}, \mathbf{i})))].$$

**Conservation of Region Information.**   The second constraint is that given a region $\mathbf{i}$ generated from a latent vector $\mathbf{z_i}$, the final image $\mathtt{G_F}(\mathbf{I}, \mathbf{z_i}, \mathbf{i})$ must contain information about $\mathbf{z_i}$. This constraint is designed to prevent the mask extractor $\mathtt{F}$ to produce empty regions. Indeed, if region $\mathbf{i}$ is empty, i.e. $\mathbf{M}^{\mathbf{i}}_{x,y} = 0$ for all $x, y$, then $\mathbf{z_i}$ cannot be retrieved from the final image. Equivalently, if $\mathbf{z_i}$ can be retrieved, then region $\mathbf{i}$ is not empty. This information conservation constraint is implemented through an additional term in the loss function. Let us denote $\delta_k$ a function which objective is to infer the value of $\mathbf{z}_k$ given any image $\mathbf{I}$. One can learn such a function simultaneously to promote conservation of information by the generator. This strategy is similar to the mutual information maximization used in InfoGAN. [9].

The final complete process is illustrated in Figure 1 and correspond to the following learning objectives:

$$\max_{\mathtt{G_F}, \delta} \mathcal{L}_{\mathtt{G}} = \mathbb{E}_{\mathbf{I}\sim p_{data}, \mathbf{i}\sim\mathcal{U}(n), z_i\sim p(z)} \big[ \mathtt{D}(\mathtt{G_F}(\mathbf{I}, \mathbf{z_i}, \mathbf{i})) - \lambda_z ||\delta_{\mathbf{i}}(\mathtt{G_F}(\mathbf{I}, \mathbf{z_i}, \mathbf{i})) - \mathbf{z_i}||_2^2 \big]$$
$$\max_{\mathtt{D}} \mathcal{L}_{\mathtt{D}} = \mathbb{E}_{\mathbf{I}\sim p_{data}} \big[ \min(0, -1 + \mathtt{D}(\mathbf{I}) \big] + \mathbb{E}_{\mathbf{I}\sim p_{data}, \mathbf{i}\sim\mathcal{U}(n), \mathbf{z_i}\sim p(\mathbf{z})} \big[ \min(0, -1 - \mathtt{D}(\mathtt{G_F}(\mathbf{I}, \mathbf{z_i}, \mathbf{i}))) \big],$$

where $\lambda_z$ is a fixed hyper-parameter that controls the strength of the information conservation constraint. Note that the constraint is necessary for our model to find non trivial solutions, as otherwise, putting the whole image into a single region is both optimal and easy to discover for the neural networks. The final learning algorithm follows classical GAN schema [3, 17, 39, 57] by updating the generator and the discriminator alternatively with the update functions presented in Algorithm 1.

Figure 1: Example generation with $\mathtt{G_f}(\mathbf{I}, \mathbf{z_i}, \mathbf{i})$ with $\mathbf{i} = 1$ and $n = 2$. Learned functions are in color.

---

**Algorithm 1** Networks update functions

---

1: **procedure** GENERATORUPDATE
2:     sample data $\mathbf{I} \sim p_{data}$,
3:     sample region $\mathbf{i} \sim \mathtt{Uniform}(\{1, \ldots, n\})$
4:     sample noise vector $\mathbf{z_i} \sim p(\mathbf{z})$
5:     $\mathbf{I}_{gen} \leftarrow \mathtt{G_f}(\mathbf{I}, \mathbf{z_i}, \mathbf{i})$                             ▷ generate image
6:     $\mathcal{L}_z \leftarrow -||\delta_{\mathbf{i}}(\mathbf{I}_{gen}) - \mathbf{z_i}||_2$         ▷ compute information conservation loss
7:     $\mathcal{L}_G \leftarrow \mathtt{D}(\mathbf{I}_{gen})$                       ▷ compute adversarial loss
8:     update $\mathtt{G_f}$ with $\nabla_{\mathtt{G_f}}\left[\mathcal{L}_G + \mathcal{L}_z\right]$
9:     update $\delta_{\mathbf{i}}$ with $\nabla_{\delta_{\mathbf{i}}}\mathcal{L}_z$
10: **procedure** DISCRIMINATORUPDATE
11:     sample datapoints $\mathbf{I}_{real}, \mathbf{I}_{input} \sim p_{data}$
12:     sample region $\mathbf{i} \sim \mathtt{Uniform}(\{1, \ldots, n\})$
13:     sample noise vector $\mathbf{z_i} \sim p(\mathbf{z})$
14:     $\mathbf{I}_{gen} \leftarrow \mathtt{G_f}(\mathbf{I}_{input}, \mathbf{z_i}, \mathbf{i})$                   ▷ generate image
15:     $\mathcal{L}_D \leftarrow \min(0, -1 + \mathtt{D}(\mathbf{I}_{real})) + \min(0, -1 - \mathtt{D}(\mathbf{I}_{gen})$    ▷ compute adversarial loss

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

# Supplementary Material for Unsupervised Object Segmentation by Redrawing

We provide architectural details, hyper-parameters discussion and additional samples output of our model.

## Architectural details

| mask network $\mathtt{f}(\mathbf{I})$ | nonlinearities | output size |
| --- | --- | --- |
| Image $\mathbf{I}$ | | 3x128x128 |
| Conv 7x7 (reflect. pad 3) | Instance Norm, ReLU | 16x128x128 |
| Conv 3x3 (stride 2, pad 1) | Instance Norm, ReLU | 32x64x64 |
| Conv 3x3 (stride 2, pad 1) | Instance Norm, ReLU | 64x32x32 |
| Residual Bloc (Instance Norm, ReLU) | | 64x32x32 |
| Residual Bloc (Instance Norm, ReLU) | | 64x32x32 |
| Residual Bloc (Instance Norm, ReLU) | | 64x32x32 |
| Pyramid Pooling Module | | 68x32x32 |
| Upsample | | 68x64x64 |
| Conv 3x3 (pad 1) | Instance Norm, ReLU | 34x64x64 |
| Upsample | | 34x128x128 |
| Conv 3x3 (pad 1) | Instance Norm, ReLU | 17x128x128 |
| Conv 3x3 (reflect. pad 3) | sigmoid (if $n = 2$) or softmax | $n$x128x128 |

Table 2: Architecture of mask network $\mathtt{f}$. Overall architecture is similar to Cycle-GAN for image translation, except with less residual blocs but a Pyramid Pooling Module introduced by PSPNet.

| discriminator network $\mathtt{D}$ and encoder $\delta$ | output size |
| --- | --- |
| image input $\mathbf{I}$ | 3x128x128 |
| Down Res Bloc (ReLU) | 64x64x64 |
| Self-Attention Bloc | |
| Down Res Bloc (ReLU) | 128x32x32 |
| Down Res Bloc (ReLU) | 256x16x16 |
| Down Res Bloc (ReLU) | 512x8x8 |
| Down Res Bloc (ReLU) | 1024x4x4 |
| Res Bloc (ReLU) | 1024x4x4 |
| Spatial sum pooling | 1024x1x1 |
| Linear | 1 for $\mathtt{D}$, 32 for $\delta$ |

Table 3: Architecture of discriminator network $\mathtt{D}$ and encoder $\delta$.

**Hyperparameters**

We discuss some notable hyper-parameters and architectural choices. We identified $\lambda_z$ and the initialization scheme as critical hyper-parameters and focus our search on those, in addition to the standard learning rates search. The number of channels in our generators is also important.

- Learning rates are set to $10^{-4}$ except for mask network for which we use a smaller learning rate ($10^{-5}$). We searched for learning rates independently for each neural network.

- Batch size were set between 20 and 25, as much as we could fit on a single Nvidia Tesla P100 GPU.

- We chose smaller $size_z$ for each $\mathbf{z}_k$ (16 for c2-MNIST and 32 for the other datasets) than what is usually found in GAN literature so that the vectors could be reasonably retrieved by $\delta$.

- As expected, we found that $\lambda_z$ were important to tune for the stability of our training procedure. We use $\lambda_z = \frac{5}{n*size_z}$ for all datasets except LFW where $\lambda_z = \frac{15}{n*size_z}$.

- Adequate orthogonal initialization is critical. We found that our model worked best when initialized with a gain at 0.8, and not at all when set at .2 and 1.4.

- We tested smaller numbers of channels for our generators $\mathsf{G}_k$ since each generator only have to model a specific type of object. We still use the same number ($ch = 64$) as the other networks for LFW and colored-2-MNIST but reduced to $ch = 32$ for CUB and Flowers.

- We used spectral normalization for all our networks except the mask network on which a weight decay of $10^{-4}$ is applied instead.

- The use of pyramid pooling produce masks of significantly better quality that standard residual network.

- The use of self-attention in both D and G might not be mandatory but more experiments would be needed to conclude.

- We found that having different ratio of discriminator updates and generator updates didn't help.

**Additional output masks**

Figure 5: Masks obtained for images from the Flowers test set. Each bloc of three rows depict from top to bottom input image, ground truth and output of our model. Each bloc from top to bottom: 1) top masks according to accuracy 2) top masks according to IoU 3-4) randomly sampled masks 5) worst masks for accuracy 6) worst masks for IoU. Because the ground truth for Flowers were obtained via an automated process, our model actually provide better predictions in worst agreement case.

Figure 6: Masks obtained for images from the CUB test set. Each bloc of three rows depict from top to bottom input image, ground truth and output of our model. Each bloc from top to bottom: 1) top masks according to accuracy 2) top masks according to IoU 3-4) randomly sampled masks 5) worst masks for accuracy 6) worst masks for IoU.

Figure 7: More randomly sampled output masks on Flower dataset.

Figure 8: More randomly sampled output masks on CUB dataset.