[Reviews · NeurIPS 2019]

Reviewer 1



Originality: To my knowledge this is an original approach to the unsupervised learning of object segmentation. Besides the specific method proposed in this paper, I find this problem very exciting and I think that it will have a great development in the near future. I do not see this as a combination of prior work. In general, this paper uses methodologies/toola that are becoming well-established (eg GANs). There is a good prior work section, but some important works are missing and should be discussed, such as: Remez et al. Learning to segment via cut-and-paste. ECCV 2018. Ostyakov et al. SEIGAN: towards compositional image generation by simultaneously learning to segment, enhance, and inpaint, ArXiv 2018 Kanezaki. Unsupervised image segmentation by backpropagation. ICASSP 2018 Ji et al. Invariant information clustering for unsupervised image classification and segmentation. ArXiv 2018. In any case, this paper proposes a different method compared to those in the missing citations. ******************************************** Quality: The contribution is mostly on a solution to a very challenging problem. Therefore, there is more of an algorithmic and experimental contribution. In general the approach is in the right direction. However, there are some technical issues that I find unconvincing and I would like the authors to explain them. 1) The independence assumption between the textures in each layer is not true (the authors also mention this at lines 137-140). I expect that this limits the applicability of the method to specific datasets where there is a single object category or where background and foreground are mostly independent (so the context is irrelevant). If this is the case, I find the whole idea limited as it could not be, in principle, applied to datasets with a mix of different categories. 2) The other important restriction in this method is the mask constraint: Basically, only objects with the same mask can be generated. This could limit the variability of the generated objects. 3) Another potential problem is the use of a regressor of z_k. The idea is technically correct, but the generator could easily "fool" this strategy. It could learn to vary the foreground object only a little bit, enough for the delta_k function to retrieve the latent code z_k. Since we know that adversarial examples can change the classification result with an imperceptible change of the input, we could also expect the generator to do the same. Besides, producing a small variation is certainly easier than a large one, so it is more likely to behave in this undesired way. 4) It is also unclear why the training would not simply result in an f function that outputs the same set of masks, where there is a single foreground object (M1=1 and M2=0 everywhere). Then, G1 is simply a GAN generator. I could not see a term that would discourage this degenerate behavior from happening. Probably it would be good to see an analysis of and convincing experiments on these issues. For example, what happens when 2 categories are mixed. Another one is how much variability is possible for the same mask. If not a quantitative evaluation, it would be good to see a qualitative one. An ablation showing the effect of using or not using delta_k would be useful. An explanation of why f does not output a content would be very useful. ******************************************** Clarity: The paper is clearly written and well organized. I think that the only missing components are the points above, The code seems reproducible. ******************************************** Significance: In my view unsupervised object segmentation is a very important problem. The results in this paper are a first step. In my view their main restriction is that they can only work with a single category at a time, but this is still a step forward. However, technically this should be called weakly supervised object segmentation, as the category label would be needed. Experimentally the conclusions are that the method yields a meaningful segmentation. However, my doubts above are still waiting for a clarification from the authors. ++++++++++++++++++++++++++ After reading all the reviews and the author's rebuttal we discussed briefly. I decided to keep my score.

Reviewer 2



The idea of training using an adversarial approach, and generating region by region, instead of the whole image, goes towards building robustness on region segmentation. Results seems to demonstrate the effectiveness of this novel approach. The assumption of the paper, that training on partly reconstructed images should yield as good results than training on whole new images, seems intuitive. This is the first work that I aware of, tacking this approach.

Reviewer 3



Originality: The proposed framework for unsupervised object segmentation is novel and its the first work using generative models to demonstrate unsupervised segmentation on real world datasets. The design of the segmentation model and the learner is done by combining ideas from prior work. Quality: The paper is technically sound, and the experimental results show that proposed model performs well on three datasets showing the effectiveness of the approach. However, the evaluation is done in settings with a single fore-ground object, while the described model is for "n" different objects. Clarity: The paper is well written and experimental details are clearly mentioned. Significance: The proposed unsupervised segmentation model is shown to be effective on simple single object images. This is a step-up from recent prior work~[14][18], which is largely demonstrated on synthetic datasets. Future work could build on these ideas to extend this to segment varying number of objects on real images. ------------------------------- Post-rebuttal --------------------------------------- I commend the authors for a well written rebuttal. The experiments with the combined LFW+Flowers dataset was interesting and illustrates that method works with more diverse real image datasets, which is a step beyond recent prior work. I increase my score to 7 and recommend accepting the paper.

[Author Response · NeurIPS 2019]

We thank the reviewers for their relevant and insightful feedback. We are also grateful for pointing out related work
we have missed and we will include them in the final version of the paper. Will also be included additional comments
formulated in this response page and more data points will be added to Figure 3 (relative to the performance of
supervised baselines) following recommendations of Reviewer 3. Unfortunately, the updated figure and associated
commentary had to be cut from this response.

**[R-1 issue-1] Does the independence assumption hurt the general usefulness of our method?**
In our framework, the independence between regions is conditional to a mask. Therefore, our generator can infer some
information about the objects, e.g. class label, from the masks. Following the recommendation of Reviewer 1, we
trained ReDO on a combination of LFW and Flowers images (without labels). This new dataset has more variability,
contains different types of objects, and display a more obvious correlation between the object and the background.
Re-using the same hyper-parameters as for the Flowers dataset, (probably not optimal for this fused dataset), we
obtained a reasonable accuracy of 0.856 and an IoU of 0.691. Masks and generation samples are displayed in Figure 1.
These preliminary results show that ReDO is not restricted to problems with a single object category. These updated
experiments will be included in the final version of the paper.

Figure 1: Results on LFW + Flowers dataset, arranged as in Figure 2 of the paper. As $\mathbf{z}$ is kept constant on a column
across all rows, we can observe that $\mathbf{z}$ codes for different textures depending on the class of the image even though the
generator is never given this information explicitly.

14
**[R-1 issue-2] Does our model produce a variety of objects?**
The variability of the generated images is not the goal of the model and only serves as a means to recover meaningful
masks. In principle, for our method to work, it should be sufficient that the variability of the pixel-values is non-zero.
Anyway, experiments show that samples are quite diverse even with a fixed mask, as illustrated in Figure 2 of the
submitted paper (and Figure 1 of this response page). In any case, they are diverse enough to learn to extract the masks.

**[R-1 issues-3-4, R-3 issue-3] About the role and effects of the information conservation constraint**
The reviewers noted the lack of an ablation study on the information conservation constraint. Reviewer 1 also noted
that our model can converge to a trivial solution. Indeed, without the information conservation loss, an optimal solution
to our objective function is for our system to output one mask that covers the whole image while the others are empty,
for instance, $\mathbf{M}_1 = 1, \mathbf{M}_2 = 0$.
As mentioned, albeit very succinctly, in lines 178-179 of the submitted paper, we introduced the information conservation
loss as a way to prevent this undesirable solution. When generating an image, if $\mathbf{M_i} = 0$ then information from $\mathbf{z_i}$ is
lost and cannot be retrieved from the final image. Or equivalently, if $\mathbf{z}_i$ can be retrieved then $\mathbf{M}_i$ is non-zero. Enforcing
the conservation of information of all $\mathbf{z}_k$ ensures that no region is empty.
However, as reviewer 1 also noted, generators are able to encode information in imperceptible ways, so the method is
not foolproof and a mask with near-zero values is still possible. But qualitative considerations presented in Figure 2 of
the submitted paper and the fact that our model is able to discover meaningful regions when information conservation
loss is applied (and fail to do so otherwise) show empirically the effectiveness of the constraint despite its potential
issues. These discussions will be added to the final version of the paper.

**[R-2 issue-1] Why these datasets? Why not ImageNet?**
The choice of datasets is motivated by two aspects: 1) the ability to fit a GAN on the dataset and to experiment with
it, with reasonable time and computing power, 2) the availability of ground truth segmentation masks to evaluate our
model. Unlabelled ImageNet is still a difficult setup for GANs and doesn't come with ground truth segmentation masks.

**[R-3 issue-1] On the ability of the model to segment n-objects.**
We have chosen to describe the model in the general case with $n$ objects since the ReDO approach is generic and can be
used in the case with $n > 1$. This ability is illustrated qualitatively in the paper on toy dataset cMNIST with $n = 3$.
On that experiment, we had obtained an accuracy of 0.99 and an IoU ratio of 0.78 for the two objects. These results,
however, are not entirely informative about how well ReDO would perform on a real multi-object dataset. In fact,
compared to the discovery of a single object, this is clearly a more difficult setting and would require more computing
power. We are focusing on this problem as an important future step.

[Meta-Review · NeurIPS 2019]

I thank the authors for their submission. The paper presents an algorithm for unsupervised object segmentation. I strongly encourage the authors to take into account the reviewers' comments and concerns for the final manuscript, in particular regarding failure points, weaknesses and directions for future work.